# Variability Matters

**DOI:** 10.3390/ijerph18010157

**Published:** 2020-12-28

**Authors:** Maarten Jan Wensink, Linda Juel Ahrenfeldt, Sören Möller

**Affiliations:** 1Department of Epidemiology, Biostatistics and Biodemography, University of Southern Denmark, Winsløwsvej 9B, 5000 Odense, Denmark; lahrenfeldt@health.sdu.dk; 2Department of Clinical Research, University of Southern Denmark, Winsløwsvej 9B, 5000 Odense, Denmark; moeller@health.sdu.dk

**Keywords:** inequality, statistical inference, forecasting, lifespan, socioeconomic status, academic performance

## Abstract

Much of science, including public health research, focuses on means (averages). The purpose of the present paper is to reinforce the idea that variability matters just as well. At the hand of four examples, we highlight four classes of situations where the conclusion drawn on the basis of the mean alone is qualitatively altered when variability is also considered. We suggest that some of the more serendipitous results have their origin in variability.

“A statistician confidently waded through a river that was on average 50 cm deep. He drowned.”—Godfried Bomans

## 1. Introduction

Much of science, including public health research, focuses on means (averages). There are various reasons why this may be so. An average says something about a group as a whole, whereas variability measures the extent to which members of a group differ. In a sense, then, averages tell us what we do know, whereas variability tells us what we do not know. We naturally tend to prefer knowns over unknowns.

Means have computational advantages too. Reliable estimates of variability require larger samples than accurate estimates of means. Under the modest assumptions of the central limit theorem, we know that the sampling distribution of a mean tends to a normal distribution with the same mean as the ancestral distribution, while through the basic laws of transformations of random variables we know that the standard deviation of that sampling distribution goes down with the square root of the sample size. Thus, with limited data we often gather considerable information about the mean through the shape, location, and scale of its sampling distribution. Variability around that mean is mainly discussed in situations where uncertainty in the estimate of the population standard deviation results in additional uncertainty about the mean, as in the popular *t*-test [1].

Yet there is an increasing realization that variability matters also. We see this around economic inequalities [2], inequalities in health and lifespan across social groups [3], and we see it in the selective potential of populations, which is determined by the tails of the fitness distribution [4]. Most recently, variability in disease transmission has been important in modelling and understanding the 2020 coronavirus (COVID-19) pandemic [5,6]. Moreover, with science turning to larger datasets and increasing computational power, variability becomes tractable computationally as well.

Inequality is always interesting and important. Oftentimes public health trends are driven by changes in variability rather than in the mean, or indeed by an interaction amongst these. Variability is also the fuel of scientific research, for how else can we compare across levels of exposure to a potentially causal agent? Additionally, there is an increasing realization that not all members of a population capitalize on positive health trends to the same extent.

The purpose of the present paper is to reinforce the idea that variability matters. We do not claim that this idea is new (various good researchers occupy themselves with the topic, while there is also an increasing place for variability in the public debate), but merely wish to illustrate the variety of situations in which variability provides important additional information. To do so, we give four examples that we feel highlight the issue cogently.

## 2. An Example Regarding Gender Similarities and Differences in Academic Performance

When looking at evidence on gender similarities and differences in average test scores it is important not only to look at gender differences in means but also in variance. The hypothesis of greater male variability was originally proposed in 1894 by Havelock Ellis to explain the excess of males among both the mentally deficient and among the eminent. Ellis particularly emphasized the wide social and educational significance of the phenomenon [7]. The fact that the variance of the distribution of scores for males is greater than the variance for females will result in an excess of males in both the high and low tails of the distribution [8]. This hypothesis has for example been used to try to explain the dearth of women faculty in mathematics at Harvard [9].

The most common measure of the magnitude of gender differences in psychological research is Cohen’s *d*, calculated as the mean for males minus the mean for females, divided by the pooled within-groups standard deviation (i.e., *d* is the measure of a mean difference in units of the standard deviation (SD)) [10]. Cohen’s guidelines for interpreting these effect sizes are that *d* = 0.20 is a small effect, *d* = 0.50 is a moderate effect, and *d* = 0.80 is a large effect. Hyde interprets a *d* ≤ 0.10 as trivial [11]. Differences between males and females in average mathematical tests tend to be small or non-existent [12,13]. For instance, a Danish register study of 19,766 adolescent twins and singletons found that males had higher test scores in mathematics than females, but the effect sizes were small (*d* = 0.06–0.15) [14]. However, the magnitude of this sex difference in average test scores appears to increase with age [11,15] when the mathematical concepts require more reasoning, more spatial abilities, and more complex problem-solving [16,17].

Nevertheless, the combination of a small average difference favoring males and a greater standard deviation for males than for females could lead to a gender ratio favoring males in the upper tail of the distribution, reflecting exceptional talent [11]. Although there are few gender differences in academic ability [18], and while there are several reasons for the large sex differences in choice of professional careers, female’s tendency towards the average may result in slightly more women with an academic degree [19], but it may also result in a lower fraction of women than men who have the native intellectual capacity to do science at the highest levels [19].

## 3. An Example Regarding Trends in Pre-Conception Medication Use among Future Fathers

Paternal medication use in the months preceding conception is on the rise. Some of this trend may be due to a general rise in drug use (in turn driven by particular drugs, such as anxiolytics), while an important role is also played by paternal age. When we look at Denmark 1996–2017, we see a rise in the mean prescription count that is partially driven by an increasing mean paternal age, but also partially driven by an increase in the variability in paternal age [20]. How?

Figure 1a illustrates how mean paternal medication prescriptions increase with age (after Online Figure 2 of reference [20]). Consider two fathers of each 30 years old; from Figure 1a we see that their mean number of prescriptions would be 0.6. Now instead take two fathers, one of 20 years old and one of 40 years old. The mean age of these fathers is again 30 years, and the father aged 20 years would still be expected to have 0.6 pre-conception prescriptions. Yet the father aged 40 would be expected to have 0.8 prescriptions, giving an average of 0.7 prescriptions instead of 0.6. This emerged solely by increasing the variability, while keeping the mean constant.

Now assume that paternal age follows a normal distribution. This is not exactly true (there is a skew towards higher ages), but it suffices to make the point and enables the reader to verify the result (R code in Appendix A). Figure 1b compares the results for different trends in the variability (standard deviation) of the paternal age distribution, while the mean age increases in identical ways. When the mean continues to increase but the increase in variability stops, the increase in mean prescription count is considerably less. This because the interaction of mean and standard deviation determines the proportion of fathers of, say, 40 or 50 years old, who are the main drivers of the observed trend. Such a principle, driven by a curve that is convex to the X-axis, seems to be a general phenomenon.

## 4. An Example Regarding Variability in Length of Life as a Complement to Life Expectancy

Life expectancy is an indicator of average mortality used commonly in health sciences and demography. It expresses the average number of years a person is expected to live. Usually we refer to life expectancy at birth, but not always. For example, pension funds can be interested in life expectancy at retirement, i.e., the number of years a retiree is expected to live conditional upon surviving up to retirement.

Because it is a mean, life expectancy conceals variation in length of life, which can be substantial. Historically, life expectancy at birth is negatively correlated with lifespan variation [21], meaning that as populations live longer on average, ages at death have become more similar. Indeed, examples exist where countries could reduce lifespan inequality and increase life expectancy through addressing the same causes of death [22], suggesting a coherent policy target.

Yet, such coincidental correlations are not law. Not all socioeconomic groups in society necessarily benefit equally from life expectancy improvements. Studies looking at differences in socioeconomic status found that while life expectancy at age 30 in Denmark has improved for all income groups, lifespan variation has increased for the more disadvantage individuals, while it has decreased for the rest [23]. This implies that among the more disadvantaged individuals, there are important groups that do not benefit from the increases in life expectancy seen at the aggregate level; a nuance that would have been missed by looking at the mean alone.

In the same vein, using data from the Human Mortality Database [24], we see that a moderate rise in life expectancy at age 65 for Danish females 1970–2000, from 17 to 18 years (Figure 2a), was not accompanied by a drop in lifespan inequality. The standard deviation rose from 8.0 to 8.8 years (Figure 2a), while the coefficient of variation also increased moderately (not shown). This increase in lifespan inequality was due to a stagnation in the proportion that died before age 75 at slightly above 20% (Figure 2b), along with a solid rise in the proportion that died at age 90 and above, which rose from 16% to 24% (Figure 2b). Thus, while a share of the population did capitalize on positive health trends, an important part did not. Only after the year 2000 did we see a sharp rise in life expectancy accompanied by a drop in proportion dying before age 75, and a continuation of the increase in the proportion dying at ages above 90. This more uniform development resulted in a drop in inequality, even in absolute terms (the standard deviation), suggesting that a broader share of the population now had longer lives.

## 5. An Example Regarding Variability in and Average Consumption of Alcohol with Respect to Socioeconomic Status

One of the surprising observations obtained in cohort studies of alcohol consumption is the phenomenon that while subgroups of higher socioeconomic status in many populations have a higher average alcohol consumption, at the same time individuals from subgroups with low socioeconomic status are overrepresented both among abstainers as well as among heavy drinkers, as measured by total consumption as well as frequency of episodic heavy drinking. This phenomenon has been seen in different North European populations (e.g., Sweden [25], Finland [26], and Denmark [27]).

It is important that this phenomenon is taken into account when investigating patterns of alcohol consumption in these populations, particularly as it has been documented extensively in literature that the adverse effects of high alcohol consumption are especially common in individuals of low socioeconomic status [25,26,27]. The interplay between mean and variation could in principle be part of the explanation of this observation, as even with a lower average consumption, a larger proportion of individuals with low socioeconomic status will be in the highest risk groups with respect to alcohol consumption. However, studies comparing individuals with different levels of socioeconomic status but with the same level of alcohol consumption, document, at least in some populations, that some of the differences in risk of adverse effects persists even between individuals of equal consumption [25,26,27]. Still, the opposite directions of mean and variation are of relevance to determine size and characteristic of high-risk groups with different levels of socioeconomic status. A lower mean alcohol consumption does not seem to protect low socioeconomic groups from crossing the threshold of problematic drinking.

## 6. Discussion

However witty, the accusation levelled against statisticians in the epigraph of this article is unfair. A statistician would always begin by listing some summary statistics of the data (the river, in this case), including variance, minimum and maximum values, and perhaps some quantiles. A statistician would also have some concern with the consequences of making a wrong decision: if these consequences are grave, such as drowning, a statistician would tolerate a smaller likelihood of a wrong decision than when consequences are trivial. As a result, no statistician would set out to wade confidently through a river based solely on information about its average depth.

Nevertheless, the oversimplistic approach of considering the mean alone, and not the distribution around it, is taken all too often. We have given four examples of situations where we needed to augment the information provided by the mean with information about variability in order to draw more informed conclusions. These examples illustrated four classes of situations: (1) if variability is different between groups, means alone give little insight into the share of people that reach a certain threshold and are hence selected into some group (say, high-achieving mathematicians, first example); (2) failing to account for variability gives wrong predictions of future trends (second example); (3) increasing variability implies that not all in a population capitalize on positive health trends to the same extent (third example); and (4) means without variability do not reveal the potential extent of public health issues (fourth example). A mean alone says little about the proportion of a population that crosses a threshold, while different subgroups of a population may follow different trends. This phenomenon also seems to be evident in spread of infectious diseases, most recently the Coronavirus Disease 2019 (COVID-19), where there is large heterogeneity in disease spreading [28].

There are limits to the amount of information that a single number can contain. From that perspective, it is unsurprising that we found situations where averages alone (single numbers) were insufficient to describe relevant aspects of the situation. This immediately begs the question whether a single number like the variance or standard deviation can contain all the relevant aspects of variability. Of course, if data are described sufficiently well by a two-parameter distribution, such as the normal distribution, two numbers say it all. However, in more complex situations, we may need more. Just like an average may not always suffice, descriptions of variability in a single number may be insufficient as well depending on the question we are trying to answer. We then need to complement this information with more numbers, for example skew, which help determine which part of a population crosses a threshold, or which part of a population experience increases in life expectancy.

Piketty [2] finds the Gini coefficient (a single number between 0 and 1) insufficient to describe economic inequalities and their evolution, and instead resorts to quantiles, such as the proportion of the overall wealth possessed by the poorest half of the population, the upper 10%, the upper 5%, the upper 1%, or indeed the upper 0.1% or even 0.01%. Depending on the kind of inequality and the period that he studies, four or five of such quantiles are deemed sufficient to grasp the main elements of economic inequality and its evolution. In their approach of forecasting statistical moments of the age-at-death distribution, Pascariu et al. [29] found that using seven statistical moments tends to strike a good balance between simplicity and computational tractability. Standard statistical courses often teach the first four moments (mean, variance, skew, kurtosis), all of which can be explained intelligibly with pictures, which becomes increasingly difficult for higher moments. Variability, then, is multidimensional as well.

When an inequality is detected, the level of inequality determines whether intervention is indicated, and how. For example, the insurance element of pension systems implies that some people will accumulate more pension payments than others. Indeed, the guarantee of a monthly income is precisely the point of pension systems, so inequality in the total amount of pension received is not necessarily problematic. An issue may arise when certain subgroups can systematically expect to receive a higher pension payout for every Euro they pay into the pension system. This is a matter of some concern, as pension systems often redistribute from poor to rich. Life expectancy of the rich is higher, while in-payments tend to be proportional to monthly benefits. Thus, for every Euro paid in, the rich and educated can expect to receive much more pension than the poor and uneducated, which raises questions about fairness [30].

In the same vein, consider a simple model where every human consists of the same number of cells that are all equally prone to become malignant due to genetic mutations that all occur at the same rate [31]. In such a model, there are no differences between humans at age 0. Yet the age at which cancer occurs, or indeed whether cancer occurs at all before death by competing causes [32], tends to be greatly distinct. Here we have inequality in consequences without any inequality whatsoever in baseline conditions. Yet, we wish to address them, so we should do so as inequality in the outcome emerges. Notice how chances are even at birth, but we still feel we should mend inequality in the outcome.

## 7. Conclusions

While means are easy to communicate and interpret, the distribution around the mean often provides some of the more serendipitous results. This makes variability an important topic, and we suggest that when publishing results on means, at least a check is necessary whether differences in variability could lead to materially different conclusions. The examples given highlight situations where this might be the case.

## Figures and Tables

**Figure 1 ijerph-18-00157-f001:**
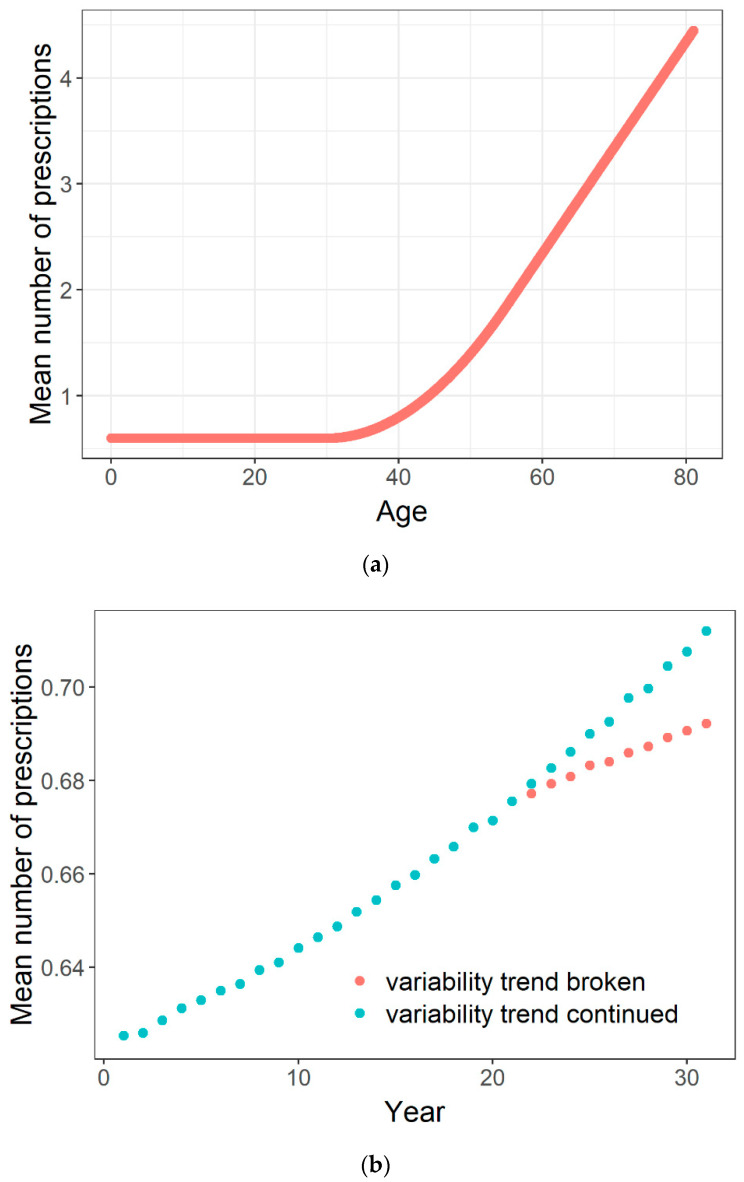
(**a**) The number of prescriptions in the six months up to conception is roughly stable up to age 30, then increases supra-linearly with age up to about age 55, after which it increases linearly with age at about 1 prescription per 10 years of age. (**b**) A simulation where both the mean and variability (standard deviation) of the age distribution increase linearly over calendar time. The pastel red dots demonstrate what would happen if the variability trend stopped at the 21^st^ year. Assumed is a normal distribution with parameters taken approximately from reference 20, with sufficient detail to prove the principle. For the turquoise dots, the mean is assumed to increase from 30 to 33 years by 0.1 year per year, while the standard deviation increases from 5 to 8 years, also by 0.1 year per year. For the pastel red dots, the mean trend is identical, but the standard deviation is kept constant at 7 from year 21 onwards.

**Figure 2 ijerph-18-00157-f002:**
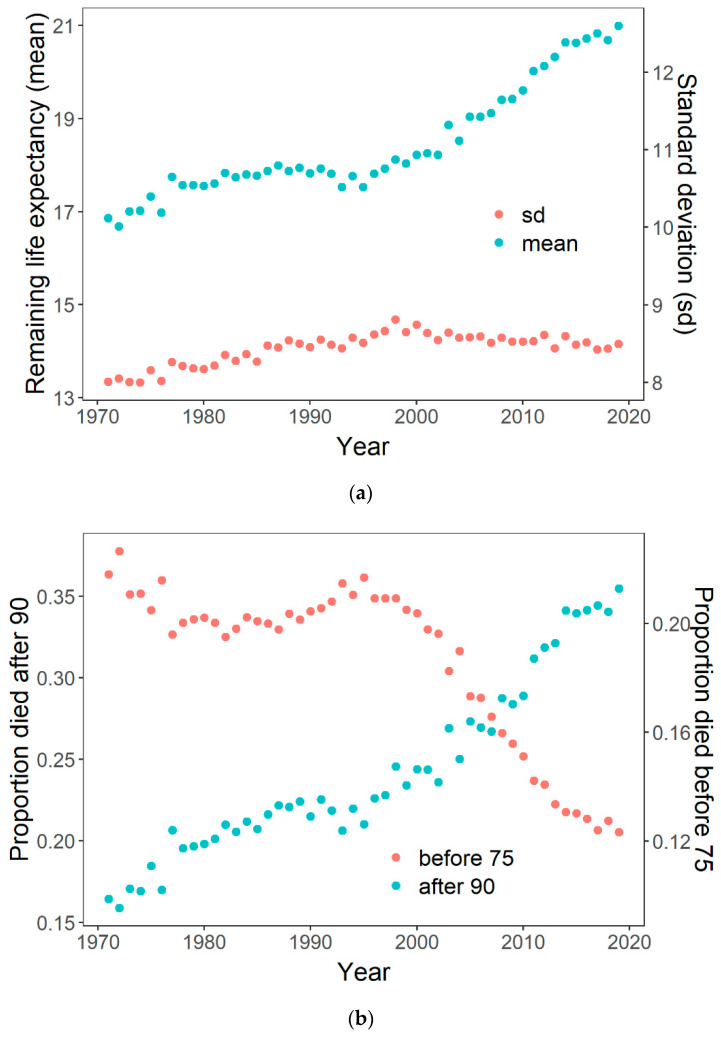
Life expectancy at age 65 for Danish females rose modestly 1970–2000, while lifespan inequality increased as well (**a**). The latter was due to the fact that a large proportion of the population continued to die before age 75, while an increasing proportion survived to higher ages (**b**).

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
