# Peer review of "Variability Matters"

_ijerph, 2020, doi:10.3390/ijerph18010157_

Round 1

Reviewer 1 Report

  • This manuscript addresses the importance of reinforcing the idea that variability matters. Four classes of examples were shown:1) when variability is different between groups, 2) wrong prediction by disregarding variability, 3) related to inequality, 4) mean with variability. The importance of variability is taken for granted. Therefore vivid examples of substantial importance are needed.
  • An example regarding gender similarities and differences in academic performance: It seems that this example showed successfully what happens when variability is different between groups.
  • An example regarding trends in pre-conception medication use among future fathers: Figure 1 shows an example of wrong prediction by disregarding variability. However 2-b), c), and d) are difficult to understand.
  • Authors need to clarify the procedure and results of wrong prediction. (I tried to run the R-program attached, but failed.) It is not easy to understand and unclear, either.

Author Response

Reviewer 1:

"This manuscript addresses the importance of reinforcing the idea that variability matters. Four classes of examples were shown:1) when variability is different between groups, 2) wrong prediction by disregarding variability, 3) related to inequality, 4) mean with variability. The importance of variability is taken for granted. Therefore vivid examples of substantial importance are needed.

An example regarding gender similarities and differences in academic performance: It seems that this example showed successfully what happens when variability is different between groups."

Reply: -We thank the reviewer for their compliment.

"An example regarding trends in pre-conception medication use among future fathers: Figure 1 shows an example of wrong prediction by disregarding variability. However 2-b), c), and d) are difficult to understand.

Authors need to clarify the procedure and results of wrong prediction. (I tried to run the R-program attached, but failed.) It is not easy to understand and unclear, either."

Reply: -We thank the reviewer for this critical comment. We agree with the reviewer that this part of the paper could be improved materially. What we tried to show in Figures 2b and 2c is the way the trend in drug prescriptions breaks away when the underlying trends in either the mean or the standard deviation of the age distribution are broken. We have critically reconsidered the way this information is presented. We have concluded that, since the topic here is variability, this example is better served with a single figure that shows the trend for continued increases in variability versus discontinued increases in variability in a single figure. We dropped the discontinuation of the mean trend, which is after all not the topic here, and may lead to confusion as the reviewer has pointed out. We hope that this accommodates the reviewer's concern regarding the figure.

This also allows us to simplify the R code. We have removed the now superfluous pieces of code, which has reduced the length of the code considerably, and added further annotation to the remainder. We hope that this solves any concerns with the R code. Figures can be made in base R, but the figures for the main manuscript are now created with ggplot2. This is the sole package that the current code relies upon.

Reviewer 2 Report

  • Authors should add more details about the implementation of the code to perform the analysis and the library involved in this task.
  • Authors should add the parameters of the methods.
  • It would be better to add some necessary arguments for Equations to make them easier to understand.
  • Please use a simple diagram or figure to illustrate the whole idea of this paper, and the modification it has been made from previous work or traditional framework.
  • Experimental results are not clear. What are the parameters used in the proposed system and how their values are set? Also, how the parameter values can affect the proposed system? Sections like Experimentation have to be extended and improved thus providing a more convincing contribution to the paper
  • What is the motivation of the proposed work? Research gaps, objectives of the proposed work should be clearly justified. The authors should consider more recent research done in the field of their study. Related works should be discussed:The different effects of BMI and WC on organ damage in patients from a cardiac rehabilitation program after acute coronary syndrome; 
  • The results of analyzes were presented in a consistent and explicit form using graphs and tables, but selection of research for analysis raises objections. Authors have proven that their model is able to generate more acceptable, but has not indicated the practical application and effects of such a solution.

Author Response

"Authors should add more details about the implementation of the code to perform the analysis and the library involved in this task."

Reply: -We thank the reviewer for this important feedback. We have shortened and simplified the code, and added important annotation. We hope that this solves any concerns with the R code. Figures can be made in base R, but the figures for the main manuscript are now created with ggplot2. This is the sole library that the current code relies upon.

"Authors should add the parameters of the methods."

Reply: -We agree. Parameters have been added to the caption of Figure 1.

"It would be better to add some necessary arguments for Equations to make them easier to understand."

Reply: -We agree. We have improved the annotation of the functions in the R code. We have added references to the figure they pertain to (if and when appropriate), so as to improve overall cohesion.

"Please use a simple diagram or figure to illustrate the whole idea of this paper, and the modification it has been made from previous work or traditional framework."

Reply: -The reviewer proposes the use of a graphical abstract. An increasing number of journals do indeed use such an abstract. We are not clear, however, if this is required or desired by the present journal. We are wholly prepared to make a graphical abstract, but we would first like to ask the editor for guidance on this matter, since this seems more of a style issue than a content issue.

"Experimental results are not clear. What are the parameters used in the proposed system and how their values are set? Also, how the parameter values can affect the proposed system?"

Reply: -We thank the reviewer for this important point. We have indeed not performed a full and thorough exploration of the parameter space, but we emphasize that we merely claim a "proof of principle". The parameter values are taken approximately from existing data, but not rigorously so, because the distribution is not an exact fit either. We take full responsibility for the confusion and have added a clarification to the caption of Figure 1.

"Sections like Experimentation have to be extended and improved thus providing a more convincing contribution to the paper What is the motivation of the proposed work? Research gaps, objectives of the proposed work should be clearly justified. The authors should consider more recent research done in the field of their study. Related works should be discussed:The different effects of BMI and WC on organ damage in patients from a cardiac rehabilitation program after acute coronary syndrome; The results of analyzes were presented in a consistent and explicit form using graphs and tables, but selection of research for analysis raises objections. Authors have proven that their model is able to generate more acceptable, but has not indicated the practical application and effects of such a solution."

Reply: -We are not entirely sure how we could satisfy these comments. For example, we are not sure how the effects of BMI and WC on organ damage from a cardiac rehabilitation program pertain to our paper. We apologize if something escaped our notice and will be happy to further improve our paper if necessary.

Round 2

Reviewer 1 Report

The manuscript is revised appropriately. Thanks for the effort.

Reviewer 2 Report

no more questions